# Sustained functional benefits after a single set of injections with abobotulinumtoxinA using a 2-mL injection volume in adults with cervical dystonia: 12-week results from a randomized, double-blind, placebo-controlled phase 3b study

Atul T. Patel[1], Mark F. Lew[2], Khashayar Dashtipour[3], Stuart Isaacson[4], Robert A. Hauser[5], William Ondo[6], Pascal Maisonobe[7], Stefan Wietek[8]*, Bruce Rubin[8], Allison Brashear[9]

1 Kansas City Bone and Joint Clinic, Overland Park, KS, United States of America, 2 Department of Neurology, Keck/University of Southern California School of Medicine, Los Angeles, CA, United States of America, 3 Department of Neurology/Movement Disorders, Loma Linda University, Loma Linda, CA, United States of America, 4 Parkinson's Disease and Movement Disorders Center of Boca Raton, Boca Raton, FL, United States of America, 5 University of South Florida Health Byrd Institute, Parkinson's Disease and Movement Disorders Center of Excellence, Tampa, FL, United States of America, 6 Methodist Neurological Institute, Houston, TX, United States of America, 7 Ipsen, Boulogne-Billancourt, France, 8 Formerly of Ipsen, Cambridge, MA, United States of America, 9 University of California Davis School of Medicine, Sacramento, CA, United States of America

* Stefanwietek15@gmail.com

## Abstract

Cervical dystonia (CD) is primarily treated with botulinum toxin, at intervals of $\geq$ 12 weeks. We present efficacy, patient-reported outcomes (PROs), and safety in adults with CD at the last available visit after a single set of abobotulinumtoxinA (aboBoNT-A) injections versus placebo using 500 U in a 2-mL injection volume. In this 12-week, randomized, double-blind trial, patients were $\geq$ 18 years of age with primary idiopathic CD, had a Toronto Western Spasmodic Torticollis Rating Scale (TWSTRS) total score $\geq$ 20, and TWSTRS-Severity subscale score > 10 at baseline. Patients (N = 134) were randomized (2:1) to aboBoNT-A (n = 89) or placebo (n = 45), with aboBoNT-A patients treated with 500 units (U) if toxin-naïve, and 250 to 500 U based on previous onabotulinumtoxinA dose if non-naïve. Endpoints included total TWSTRS, Pain Numeric Rating Scale (NRS-Pain; 24-hour), Treatment Satisfaction Questionnaire for Medication, and other PROs for pain, depression, and global health. Results are for the intent-to-treat population, with "Week 12" (Wk12) comprising the last available post-baseline assessment (end-of-study or early withdrawal). Mean TWSTRS total scores improved from 42.5 at baseline to 35.4 at Wk12 with aboBoNT-A and 42.4 to 40.4 with placebo (treatment difference: −4.8; 95% confidence interval [CI]: −8.5, −1.1; $p$ = 0.011). At Wk12, mean (95% CI) change from baseline in NRS-Pain was −1.0 (−1.59, −0.45) for aboBoNT-A and −0.2 (−0.96, 0.65) for placebo. AboBoNT-A demonstrated numeric improvements in other PROs. More aboBoNT-A–treated patients than patients receiving

**Data Availability Statement:** The data underlying the study's results cannot be shared publicly due to participant confidentiality. However, other researchers who provide a valid research question may request access to the data by submitting a proposal to DataSharing@Ipsen.com. Proposals will be assessed by a scientific review board.

**Funding:** Ipsen, Cambridge, MA, USA provided support for this study in the form of funding for medical writing support in accordance with current Good Publication Practice guidelines (GPP3) and salaries for PM, SW, and BR. The specific roles of these authors are articulated in the 'author contributions' section. The funder had no additional role in the study design, data collection and analysis, or decision to publish.

**Competing interests:** I, A. T. Patel, have read the journal's policy and the authors of this manuscript have the following competing interests: Speaker: Allergan, Ipsen, Merz, Revance; Research support: Allergan, Ipsen, Revance. I, M. F. Lew, have read the journal's policy and the authors of this manuscript have the following competing interests: Advisor/consultant/speaker: AbbVie, ACADIA, Acorda, Adamas, Cynapsus, Kyowa Kirin, Lundbeck, Neurocrine, Revance, Teva, US WorldMeds; Researcher: Biotie, Enterin Inc., Michael J. Fox Foundation, Parkinson Study Group, Pharm2B; I, A. Brashear, have read the journal's policy and the authors of this manuscript have the following competing interests: Consulting: Ipsen, Revance; Research support: paid to Wake Forest (institution) and conflicts were managed by Wake Forest. I, K. Dashtipour, have read the journal's policy and the authors of this manuscript have the following competing interests: Advisor/consultant, and/or speaker: AbbVie, ACADIA, Acorda, Adamas, Allergan, Amneal, Impax, Ipsen, Lundbeck, Neurocrine, Revance, Sunovion, Teva, US WorldMeds. I, S. Isaacson, have read the journal's policy and the authors of this manuscript have the following competing interests: Honoraria for CME/consultant/promotional speaker: AbbVie, ACADIA, Acorda, Adamas, Addex, Allergan, Amarantus, Axovant, Biogen, Britannia, Eli Lilly, Enterin, GE Healthcare, Global Kinetics, Impax, Intec Pharma, Ipsen, Kyowa, Lundbeck, Michael J. Fox Foundation, Neurocrine, Neuroderm, Parkinson Study Group, Pharma2B, Roche, Sanofi, Sunovion, Teva, UCB, US WorldMeds, Zambon; Honoraria for research grants: AbbVie, ACADIA, Acorda, Adamas, Addex, Allergan, Amarantus, Axovant, Biogen, Britannia, Eli Lilly, Enterin, GE Healthcare, Global Kinetics, Impax, Intec Pharma, Ipsen, Kyowa, Lundbeck, Michael J. Fox Foundation, Neurocrine, Neuroderm, Parkinson Study Group, Pharma2B,

placebo reported being at least "somewhat satisfied" with treatment (60.4% vs 42.2%, respectively), symptom relief (57.0% vs 40.0%), and time for treatment to work (55.8% vs 33.3%). No new adverse events were reported. Results indicate that in patients with CD, treatment with aboBoNT-A using a 2-mL injection provided sustained improvement in the TWSTRS total score and patient-perceived benefits up to 12 weeks.

**Trial registration:** Clinicaltrials.gov Identified: NCT01753310.

## Introduction

Cervical dystonia (CD), a chronic neurologic movement disorder characterized by sustained or repetitive involuntary contractions of the neck muscles that leads to abnormal postures, represents the most common focal dystonia. The prevalence of CD worldwide has been estimated to range from 20 to 4,100 cases per million [1]. Intramuscular injection of botulinum neurotoxin into affected muscles represents the primary treatment for CD [2]. Botulinum neurotoxin type A binds to receptors on peripheral cholinergic nerve endings and is internalized by receptor-mediated endocytosis. This is followed by cleavage of SNAP25, a protein needed for vesicle fusion to the presynaptic membrane, which is required for synaptosomal release of acetylcholine into the neuromuscular junction. The efficacy and safety of abobotulinumtoxinA (aboBoNT-A) for the treatment of CD has been reported based on two randomized, controlled trials and their open-label safety extensions [3–5]. AboBoNT-A is approved in the United States of America (US) and Europe for the treatment of adults with CD [6, 7].

Initially, the prescribing information in the US specified a 500-unit (U), 1-mL dilution regimen for AboBoNT-A administration; however, feedback obtained from scientific experts, community injectors, and market research studies favored use of a 500 U, 2-mL dilution. This was most likely related to prior familiarity in using comparable volumes with other approved toxins in the US.

The 2-mL dilution regimen has since been approved in the US based on the results of a 12-week, phase 3b, randomized, double-blind, placebo-controlled clinical trial that was conducted to evaluate the efficacy and safety of a 500 U, 2-mL dilution of aboBoNT-A versus placebo in adults with CD. AboBoNT-A improved the Toronto Western Spasmodic Torticollis Rating Scale (TWSTRS) total score at Week 4. AboBoNT-A demonstrated significant improvements in symptoms in both toxin-naïve and previously treated patients; a safety profile similar to the 1-mL dilution was observed [8]. An open-label extension study (NCT01753336) provided long-term safety and efficacy data, which also supported the US approval of the 2-mL dilution regimen. This manuscript expands on the 4-week results of the previously published placebo-controlled trial by providing exploratory data for up to 12 weeks to assess the durability of response with aboBoNT-A 500 U using a 2-mL dilution. This study was registered on ClinicalTrials.gov (NCT01753310).

## Methods

Subjects were given a full explanation, in lay terms, of the aims of the study, the benefits, potential discomforts and risks of taking part in the study prior to enrollment. A written explanation was also provided and written informed consent was obtained prior to enrollment. The study protocol, subject information leaflet, and informed consent document were reviewed and approved by Harrison IRB (London, OH) as well as by institutional review boards at individual study sites (if locally required) prior to commencement of the study."

Roche, Sanofi, Sunovion, Teva, UCB, US WorldMeds. I, R. A. Hauser, have read the journal's policy and the authors of this manuscript have the following competing interests: Consulting: AbbVie, Academy for Continued Healthcare Learning, ACADIA, Acorda, Adamas, AstraZeneca, Back Bay Life Science, Biotie, Bracket, Cerecor, ClearView Healthcare Partners, ClinicalMind Medical and Therapeutic Communications, Cowen and Company, Cynapsus Therapeutics, Decision Resources Group, Eli Lilly, eResearch Technology, Expert Connect, Extera Partners, GE Healthcare, Gerson Lehrman Group, Globe Life Sciences, Guidepoint Global, Health Advances, Health and Wellness Partners, HealthLogix, Huron Consulting Group, Impax, Intec Pharma, Jazz Pharmaceuticals, Kyowa Kirin Pharmaceutical Development, LCN Consulting, LifeMax, The Lockwood Group, Lundbeck, MEDACorp, Medscape, Medtronic, Michael J. Fox Foundation, Movement Disorder Society, National Institutes of Health (NIH), Neurocrine Biosciences, Neuroderm, Neuropore Therapies, Outcomes Insights, Parkinson Foundation, Peerview Press, Pennside Partners, Pfizer, Pharma2B, Phase Five Communications, Piper Jaffray & Co, Prexton Therapeutics, Projects in Knowledge, Putnam Associates, Quintiles, RMEI Medical Education for Better Outcomes, Sarepta Therapeutics, Schlesinger Associates, Scion Neurostim, Seagrove Partners, Slingshot Insights, Sun Pharma, Sunovion, Teva, US WorldMeds, Vista Research, WebMD, Windrose Consulting Group; Research support: AbbVie, Acorda Therapeutics, AstraZeneca, Axovant Sciences, Biogen, Cavion, Dart NeuroScience, Enterin, F. Hoffman-La Roche, Impax, Intec Pharma, Jazz Pharmaceuticals, Lundbeck, Michael J. Fox Foundation, NeuroDerm, Parkinson's Foundation, Prexton Therapeutics, Revance, Sunovion. I, W. Ondo, have read the journal's policy and the authors of this manuscript have the following competing interests: Advisor/ consultant/speaker: ACADIA, Acorda, Adamas, Jazz Pharmaceuticals, Neurocrine, Teva, UCB, US WorldMeds; Research support: Biogen, Lilly, Lundbeck, Sun Pharmaceuticals, Sunovian. I, P. Maisonobe, have read the journal's policy and the authors of this manuscript have the following competing interests: Employment: Ipsen. I, S. Wietek, have read the journal's policy and the authors of this manuscript have the following competing interests: Employment: Ipsen. I, B. Rubin, have read the journal's policy and the authors of this manuscript have the following competing interests: Employment (former): Ipsen.

This study was conducted under the provisions of the Declaration of Helsinki, informed consent regulations and in accordance with the International Conference on Harmonisation (ICH) Consolidated Guideline on Good Clinical Practice (GCP) and local site review boards. This study was registered with ClinicalTrials.gov (identifier: NCT01753310).

## Study design

This was a 12-week, phase 3b, multicenter, randomized, double-blind, placebo-controlled study, conducted at 43 initiated sites in the US, 38 of which enrolled patients (**S1 Fig**). Patients were randomized (2:1) to receive aboBoNT-A or placebo by intramuscular injection, stratified by whether the patient had previous treatment with onabotulinumtoxinA (onaBoNT-A). Toxin-naïve patients who were randomized to the aboBoNT-A group received 500 U, using the 2-mL dilution method in at least two affected neck muscles; patients with prior ona-BoNT-A exposure (non-naïve at baseline) received 250 to 500 U, using the 2-mL dilution method (based on previous onaBoNT-A dose) into previously injected muscles.

## Patients

Eligible patients included adults (aged $\geq$ 18 years) with a primary diagnosis of idiopathic CD (duration $\geq$ 9 months) and with a TWSTRS total score $\geq$ 20 and a TWSTRS severity subscale score > 10 at baseline. Patients could be BoNT-A naïve or could have received any other formulation of BoNT-A, including aboBoNT-A, prior to study enrollment as long as their last two injections were onaBoNT-A and they had a satisfactory clinical response to both injections.

## Treatment

All patients received a single set of intramuscular injections of either aboBoNT-A (500 U/2 mL) or placebo in a minimum of two clinically affected neck muscles. The sites and the dose per site were determined by the investigator according to the standard practice and disease presentation. Electromyography-guided injections were allowed at the preference of the investigator at each site. Patients randomized to the aboBoNT-A group received 500 U of abo-BoNT-A if they were onaBoNT-A treatment-naïve or 250–500 U of aboBoNT-A at a 2.5:1 (aboBoNT-A:onaBoNT-A) ratio to their previous onaBoNT-A dose into muscles injected during the last two sequential cycles of onaBoNT-A within the past 18 months for the treatment of CD. The amount of aboBoNT-A injected into the sternocleidomastoid (SCM) muscle was limited to $\leq$ 0.6 mL (150 U), in order to reduce the occurrence of dysphagia, per the aboBoNT-A US prescribing information [6]. Prior to administration, aboBoNT-A and placebo vials were reconstituted at the investigational site with 2-mL preservative-free sodium chloride for injection (0.9%). Detailed instructions were provided for the volume that needed to be withdrawn from the reconstituted aboBoNT-A vials.

## Assessments

The primary efficacy endpoint was change from baseline in TWSTRS total score at Week 4, with secondary endpoints in the testing hierarchy comprising TWSTRS total score, TWSTRS response, Patient Global Impression of Change (PGIC), and the Cervical Dystonia Impact Profile-58 (CDIP-58) at Week 2 and Week 4. Tertiary efficacy endpoints included assessments of the change from baseline at Week 12.

The TWSTRS total score comprises three subscale scores: severity, disability, and pain, each of which is scored independently and can have a value from 0 to 85 (best to worst) [9]. The

TWSTRS total score was used to assess the severity of CD and was assessed by the investigator prior to study treatment at baseline (Day 1) and at all post-treatment visits. PGIC was assessed using a 7-point Likert scale, ranging from +3 (very much improved) to –3 (very much worse). The Pain Numeric Rating Scale (NRS-Pain) is a numeric, 24-hour assessment scale that measures pain as 0 (no pain) to 10 (worst possible pain). The Brief Pain Inventory (BPI) short form was used to assess the effect of CD pain on seven areas (general activity, mood, walking ability, normal work, relations with other people, sleep, and enjoyment of life) and ranges from 0 (does not interfere) to 10 (completely interferes).

## Sample size

It was determined that 132 randomized patients would be sufficient to demonstrate the superiority of aboBoNT-A to placebo assuming a minimum clinically relevant difference in the adjusted least squares mean change from baseline in TWSTRS total score at Week 4 between aboBoNT-A 2-mL and placebo equal to 5.5, a common standard deviation (SD) in the change from baseline in TWSTRS total score at Week 4 equal to 8.8, a power of 90%, a two-tailed type I error equal to 5%, and 10% dropout rate.

## Randomization

The sponsor's randomization manager, who was a statistician independent from the study, prepared two lists that were performed in blocks and were based on a computer-generated randomization list. The first list of randomization numbers stratified for subjects who were BoNT-A treatment naïve or non-naïve at baseline was generated with a 2:1 ratio of aboBoNT-A:onaBoNT-A. The second was a list of treatment numbers, which were specified on the treatment packs, to be dispatched to the sites in order to dispense the drug. This list was produced on a 1:1 basis (aboBoNT-A:onaBoNT-A). The randomization, as well as the treatment number assignation at drug dispensation, was managed by an Interactive Web Response System (IWRS) service. At screening, potential patients were assigned a patient number. Following confirmation of eligibility, patients were given a randomization/treatment allocation number and were assigned to one of the two groups (aboBoNT-A or placebo) at baseline, in sequential order within each site (and within each level of strata). This was a single-dose study. The sites of injection and the dose per site were determined by the investigator according to the standard practice and disease presentation.

## Blinding

Patients and investigators were kept blinded to the allocation. Study treatments were similar in size, color, smell, and appearance, allowing the blinded conditions of the study to be maintained. A set of individually sealed code-break envelopes was prepared by the sponsor's randomization manager to enable emergency code-break procedures of individual patients without compromising the blinding of the study and was provided to the Central Department of Pharmacovigilance at Ipsen.

## Statistical analysis

The statistical analysis of the efficacy and safety data was performed using Statistical Analysis Software (SAS; version 9.2). The intent-to-treat (ITT) population included all randomized patients; the modified ITT population included randomized patients with both a baseline and Week 4 post-treatment TWSTRS total score assessment. Efficacy results are presented for the ITT population (all randomized patients). The last available post-baseline assessment was either

the end of study (Week 12) or early withdrawal. Withdrawals included patients who exited the study early and patients who began the open-label protocol due to lack of efficacy after 4 weeks.

For TWSTRS total and subscores, the change from baseline is expressed as weighted overall treatment difference and 95% confidence interval (CI). Significance testing for the superiority of aboBoNT-A versus placebo was conducted for the primary and secondary efficacy endpoints at a two-tailed significance level of 5% by using a stratified analysis of covariance (ANCOVA) with baseline TWSTRS total score as covariate and stratified by the randomization stratification factor. The hierarchical testing procedure was stopped if there was no statistically significant treatment effect on the primary efficacy endpoint. Otherwise, the superiority of aboBoNT-A was then tested for each secondary endpoint in rank order, proceeding to the next endpoint only when treatment effect on the current endpoint was deemed statistically significant. Statistics were summarized by treatment group for tertiary endpoints but were not compared by formal statistical testing.

### Data sharing

Where patient data can be anonymized, Ipsen will share all individual participant data that underlie the results reported in this article with qualified researchers who provide a valid research question. Study documents, such as the study protocol and clinical study report, are not always available. Proposals should be submitted to DataSharing@Ipsen.com and will be assessed by a scientific review board. Data are available beginning 6 months and ending 5 years after publication; after this time, only raw data may be available.

## Results

### Patient disposition and study exposure

A total of 134 patients were randomized and included in the ITT population (aboBoNT-A, n = 89; placebo, n = 45) (**Table 1**). One patient was excluded from the analysis because he did not receive study drug. From the aboBoNT-A group 87 patients (98%) attended Week 2 visits, 84 (94%) attended Week 4, and 57 (64%) attended Week 12. In the placebo group 45 patients (100%) attended Week 2 visits, 45 (100%) attended Week 4, and 21 (47%) attended Week 12 (**Fig 1**). Forty-seven patients (aboBoNT-A, n = 25 [28%]; placebo, n = 22 [49%]) elected to enter the open-label portion of the study prior to Week 12, with 33 entering the open-label study at Week 4. **Table 2** shows a post-hoc analysis of patient study exposure based on all ITT patients and all of those who withdrew from the study early.

Overall study duration took place from January 2013 to January 2015, which included patient recruitment (approximately 9 months) and the last patient follow-up (up to 4 months). Individual treatment duration was between 4 weeks and 16 weeks, due to the study design, which allowed a ±2 day window to complete the Week 2 and 4 visits, and a 28 day window to complete the Week 12 visit.

### TWSTRS

Previously reported mean (SD) TWSTRS total score at baseline was 42.5 (10.40) for aboBoNT-A and 42.4 (10.63) for placebo; significant decreases in TWSTRS total score with aboBoNT-A versus placebo were observed at Week 2, with further improvements at Week 4 [8]. For the last available data at Week 12 or at early study withdrawal, the weighted overall treatment difference between aboBoNT-A and placebo was also statistically significant (–4.8; 95% CI: –8.5, –1.1; $p$ = 0.011; **Fig 2**). As tertiary endpoints, the TWSTRS subscales were not compared by formal statistical testing; however, at all of the time points, aboBoNT-A treatment

**Table 1. Baseline demographics and characteristics (ITT populations).**

| | aboBoNT-A | Placebo |
|---|---|---|
| | n = 89 | n = 45 |
| **Female sex, *n* (%)** | 59 (66.3) | 28 (62.2) |
| **Age (years; mean ± SD)** | 57.3 ± 11.11 | 56.5 ± 11.74 |
| **Caucasian/white race, *n* (%)** | 84 (94.4) | 42 (93.3) |
| **Type of CD, *n* (%)** | | |
| Torticollis | 75 (84.3) | 39 (86.7) |
| Laterocollis | 54 (60.7) | 30 (66.7) |
| Anterocollis | 17 (19.1) | 7 (15.6) |
| Retrocollis | 20 (22.5) | 8 (17.8) |
| Lateral shift | 22 (24.7) | 14 (31.1) |
| Sagittal shift | 9 (10.1) | 8 (17.8) |
| **Previous exposure to botulinum neurotoxin, *n* (%)** | | |
| Yes | 57 (64.0) | 29 (64.4) |
| No | 32 (36.0) | 16 (35.6) |
| **Previous botulinum neurotoxin treatments for CD, *n* (%)** | | |
| *n**[*] | 57/57 (100.00) | 29/29 (100.0) |
| onaBoNT-A | 56/57 (98.2) | 28/29 (96.6) |
| aboBoNT-A | 2/57 (3.5) | 1/29 (3.4) |
| incoBoNT-A | 6/57 (10.5) | 2/29 (6.9) |
| rimaBoNT-B | 5/57 (8.8) | 2/29 (6.9) |
| **Most recent CD treatment with onaBoNT-A**[**] | | |
| *n* | 55 | 25 |
| Mean dose[***] (U; mean ± SD) | 177.0 ± 34.0 | 176.2 ± 42.2 |
| Mean time since last injection (months; mean ± SD) | 4.8 ± 19.1 | 3.6 ± 16.3 |

[*]Subjects may have been previously treated with ≥1 botulinum neurotoxin.

[**]In subjects who were non-naïve for onaBoNT-A treatment.

[***]Mean dose of onaBoNT-A exceeds 200 U eligibility requirement due to protocol deviations, patients were excluded from ITT.

aboBoNT-A = abobotulinumtoxinA; CD = cervical dystonia; incoBoNT-A = incobotulinumtoxinA; ITT = intent-to-treat; onaBoNT-A = onabotulinumtoxinA; rimaBoNt-B = rimabotulinumtoxinB; SD = standard deviation.

demonstrated numerically greater reductions in the TWSTRS-Severity, TWSTRS-Pain, and TWSTRS-Disability subscale scores.

## Patient-reported outcomes

With the exception of CDIP-58 at Week 2 and Week 4, patient-reported outcomes (PROs) were tertiary endpoints and were not compared by formal statistical testing. At all of the time points, the majority of aboBoNT-A–treated patients indicated minimal improvements or greater via PGIC scores, at a rate approximately twice that of placebo (**Fig 3A**). More patients receiving placebo than aboBoNT-A–treated patients indicated no change or worse than at baseline. At Week 4, 34.1% and 31.8% of patients receiving placebo reported no change or worsening, respectively. For a more detailed analysis of PGIC at each time point, see **S3 Fig**. In the patient-reported rating of pain reflecting the previous 24 hours, the mean change from baseline in NRS-Pain was Δ –1.0 (95% CI: –1.59, –0.45) for aboBoNT-A and Δ –0.2 (95% CI: –0.96, 0.65) for placebo, with a consistent reduction from baseline at Week 2 and Week 4 for aboBoNT-A–treated patients, indicating reduced pain (**Fig 3B**). AboBoNT-A also

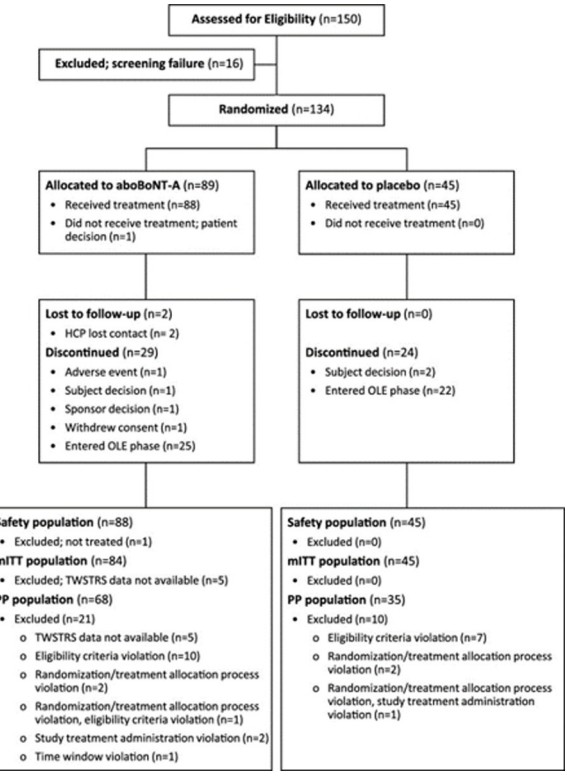

**Fig 1. CONSORT diagram.** aboBoNT-A = abobotulinumtoxinA; HCP = healthcare provider; mITT = modified intent-to-treat; OLE = open-label extension; PP = per protocol; TWSTRS = Toronto Western Spasmodic Torticollis Rating Scale. *Entered NCT01753336 (OLE) to continue active treatment.

demonstrated numeric benefits on the BPI assessment (**Fig 3C**), although the trajectory was similar in both treatment groups.

For aboBoNT-A–treated patients, there was a consistent shift toward less depression from baseline to Week 2 and continuing through Week 4 and the last available assessment (**Fig 3D**).

**Table 2. Patient (ITT) study exposure for all patients and patients who withdrew from the study.**

| ITT Population | | | |
|---|---|---|---|
| Parameter | AboBoNT-A | Placebo | Total |
| N | 89 | 45 | 134 |
| Mean (SD) days | 67.7 (38.4) | 52.5 (34.7) | 62.6 (37.8) |
| CI 95% | 59.6; 75.8 | 42.1; 62.9 | 56.2; 69.1 |
| Median days | 85.0 | 35.0 | 85.0 |
| Min; Max | 1.0; 184.0 | 14.0; 105.0 | 1.0; 184.0 |
| **ITT Patients Who Withdrew From Study** | | | |
| N | 32 | 24 | 56 |
| Mean (SD) days | 19.8 (8.4) | 22.0 (8.6) | 20.7 (8.5) |
| CI 95% | 16.7; 22.8 | 18.4; 25.6 | 18.4; 23.0 |
| Median days | 15.5 | 18.0 | 16.0 |
| Min; Max | 1.0; 37.0 | 14.0; 45.0 | 1.0; 45.0 |

aboBoNT-A = abobotulinumtoxinA; CI = confidence interval; ITT = intent-to-treat; SD = standard deviation.

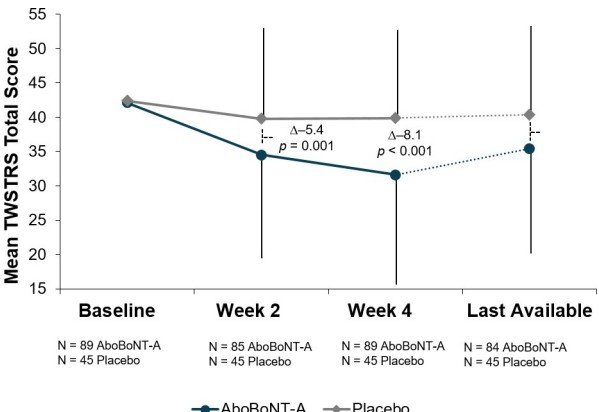

**Fig 2. Mean TWSTRS total score.** aboBoNT-A = abobotulinumtoxinA; ITT = intent-to-treat (all randomized patients); last available* = last available post-baseline (end of study or early withdrawal); TWSTRS = Toronto Western Spasmodic Torticollis Rating Scale; Δ = weighted overall treatment difference. Error lines indicate standard deviation. *Last available = last available post-baseline (end of study or early withdrawal), mean (SD) study drug exposure: 62.6 (37.8) days.

At Week 2, 59.4% of aboBoNT-A–treated patients had no or minimal depression compared with 47.3% of patients receiving placebo. At Week 4, the rates were 63.0% and 48.6% for abo-BoNT-A and placebo, respectively. At the last available assessment, 61.0% of aboBoNT-A–treated patients had no or minimal depression compared with 46.0% of patients receiving placebo. The change from baseline in the CDIP-58–scaled sum total score was consistently greater with aboBoNT-A than with placebo (**Fig 3E**). At Week 2 aboBoNT-A was Δ –5.8 (95% CI: –8.85, –2.76) compared with placebo at Δ –4.3 (95% CI: –8.79, 0.13). At Week 4 aboBoNT-A was Δ –8.5 (95% CI: –11.61, –5.30) compared with placebo at Δ –4.7 (95% CI: –9.60, 0.24) ($p$ = not significant). At the last available assessment, aboBoNT-A was Δ –7.9 (95% CI: –11.01, –4.70) compared with placebo at Δ –3.1 (95% CI: –7.75, 1.45) ($p$ = not significant).

## Satisfaction

Across time points, approximately 65% of aboBoNT-A–treated patients were at least somewhat satisfied or greater with the medication overall according to results of the modified Treatment Satisfaction Questionnaire for Medication (TSQM-9). Patients receiving placebo reported lower satisfaction rates, with 32.5% at Week 2, 35.6% at Week 4, and 40.0% of patients at the last available assessment being at least somewhat satisfied or more. Regarding satisfaction with symptom relief, 54.1% of aboBoNT-A–treated patients were at least somewhat or more satisfied at Week 2, increasing to 58.1% at Week 4 and 57.0% for last available assessment. Rates for placebo decreased from Week 2 (28.0%) to Week 4 (24.4%), with 40.0% of patients receiving placebo being at least somewhat satisfied with symptom relief at the last available assessment. More than half of aboBoNT-A–treated patients were at least somewhat satisfied with the length of time it took for medication to work (Week 2, 53.0%; Week 4, 59.3%; last available, 55.8%). Only 23.3% of patients receiving placebo were at least somewhat satisfied at Week 2, decreasing to 20.0% at Week 4, and increasing to 33.3% for the last available assessment.

## Safety

The treatment-emergent adverse events (TEAEs) are reported in **Table 3** [7]. For those receiving aboBoNT-A, 41% of patients reported TEAEs, compared with 22% of patients receiving placebo. In the aboBoNT-A group, one patient discontinued the study due to the emergence

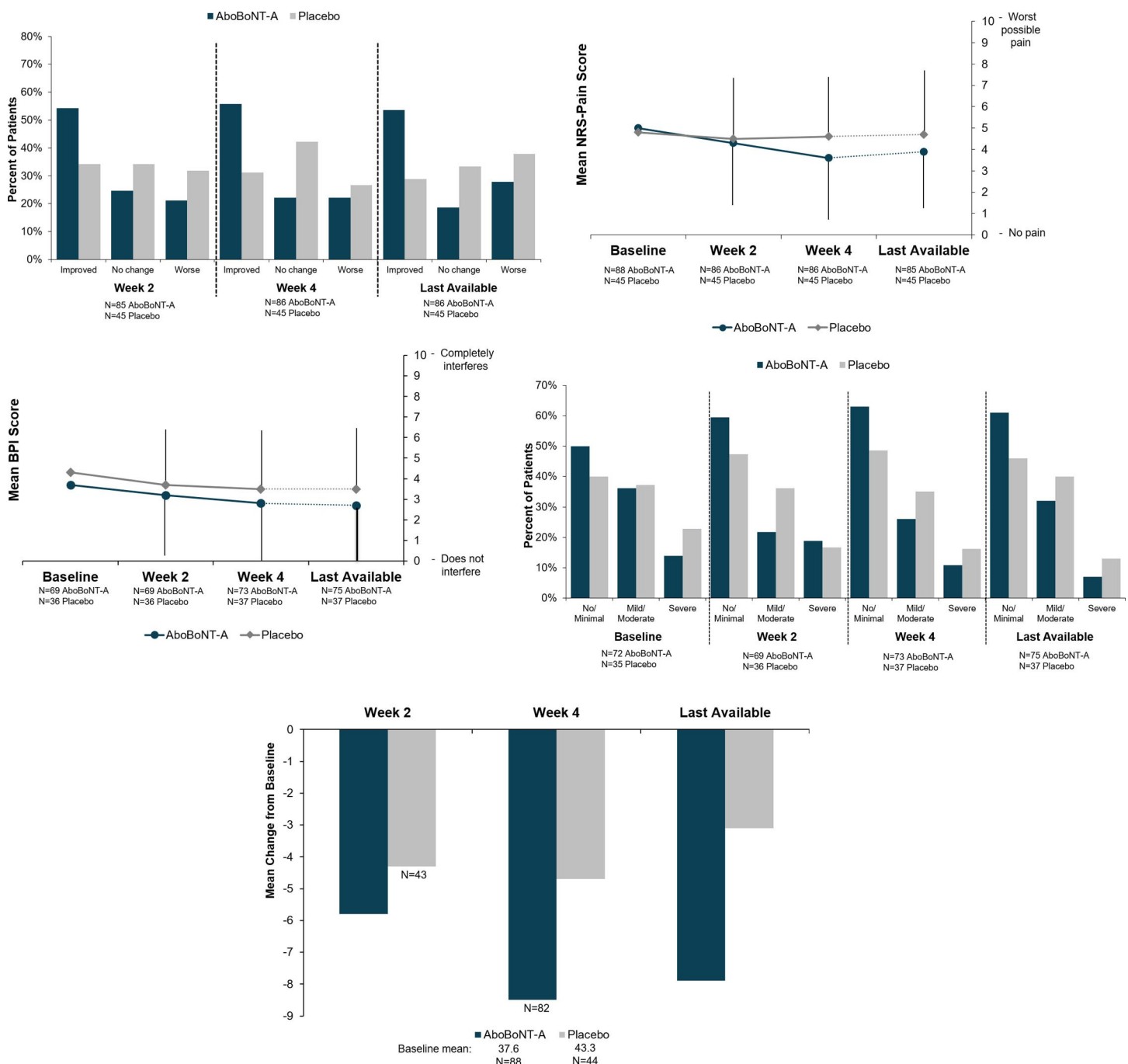

**Fig 3.** Patient-reported outcomes (ITT population): A) PGIC across time points. B) Mean NRS-Pain scores across time points. C) Mean BPI scores across time points. D) Level of depressive (PHQ-9) mood across time points. E) Change from baseline in CDIP-58 scaled sum total score across time points. BPI = Brief Pain Inventory; CDIP-58 = Cervical Dystonia Impact Profile-58; ITT = intent-to-treat (all randomized patients); Improved = minimally, much, or very much improved; Last available* = last available post-baseline (end of study or early withdrawal); Mild/moderate = mild depression (5–9) or moderate depression (10–14); No/minimal = no depression (0) or minimal depression (1–4); NRS-Pain = numeric rating scale for pain; Worse = minimally, much, or very much worse; PGIC = Patient Global Impression of Change; PHQ-9 = Patient Health Questionnaire-9; Severe = moderately severe depression (15–19) or severe depression (20–27). * Last available = last available post-baseline (end of study or early withdrawal), mean (SD) study drug exposure: 62.6 (37.8) days.

of a TEAE (colon neoplasm) not thought to be related to study medication. No patients in the placebo group discontinued the study due to an AE [8].

**Table 3. Treatment-emergent adverse events in the safety set.**

| Event, *n* (%) | aboBoNT-A | Placebo |
|---|---|---|
| | n = 88 (100.0) | n = 45 (100.0) |
| All treatment-emergent adverse events | 36 (40.9) | 10 (22.2) |
| Dysphagia | 8 (9.1) | 0 (0.0) |
| Muscular weakness | 8 (9.1) | 0 (0.0) |
| Neck Pain | 7 (8.0) | 0 (0.0) |
| Headache | 5 (5.7) | 0 (0.0) |
| Sinusitis | 3 (3.4) | 0 (0.0) |
| Bronchitis | 2 (2.3) | 0 (0.0) |
| Burning sensation | 2 (2.3) | 0 (0.0) |
| Depression | 2 (2.3) | 1 (2.2) |
| Diarrhea | 2 (2.3) | 0 (0.0) |
| Fatigue | 2 (2.3) | 0 (0.0) |
| Blurred vision | 2 (2.3) | 0 (0.0) |
| Serious Adverse Events | 4 (4.5) | 1 (2.2) |
| Dysphagia | 1 (1.1) | 0 (0.0) |
| Colon neoplasm | 1 (1.1) | 0 (0.0) |
| Endometrial cancer | 1 (1.1) | 0 (0.0) |
| Transient ischemic attack | 1 (1.1) | 0 (0.0) |
| Depression | 0 (0.0) | 1 (2.2) |

aboBoNT-A = abobotulinumtoxinA.

## Discussion/conclusion

These results demonstrate the sustained benefit of treatment with aboBoNT-A 500 U/2-mL dilution for up to 12 weeks in patients with CD, further supporting the use of the 2-mL dilution regimen and allowing dilution flexibility for clinicians.

In order to better understand how early withdrawal may have impacted the last available visit analysis, a descriptive analysis of TWSTRS at Week 12 was performed based only on patients who completed the study and those who withdrew early (with available TWSTRS data at week 12). The unadjusted difference between placebo and aboBoNT-A is 4.7 in favor of aboBoNT-A. These results are very similar to the results from the last available assessment analysis (4.8), demonstrating the benefit of aboBoNT-A at Week 12.

Efficacy of the 2-mL dilution of aboBoNT-A is similar to results from previous studies of 1-mL dilution of aboBoNT-A 500 U [3, 4]. For both of the trials using the 1-mL dilution, aboBoNT-A was significantly more effective than placebo in the change from baseline in TWSTRS score. The adjusted mean difference in the current study was −8.3 at Week 4, which is similar to the −6.0 and −8.8 observed in the pivotal clinical trial at Week 4. The weighted overall treatment difference between aboBoNT-A and placebo was −4.8 at Week 12 or at early study withdrawal in the current trial, similar to the adjusted mean difference of −4.3 and -4.1 in the pivotal clinical trials at Week 12.

The safety profile of the 500 U/2-mL aboBoNT-A dilution is similar to that observed with the 500 U/1-mL dose used in the aboBoNT-A pivotal trial. In the current study, TEAEs were reported in 41% of patients receiving aboBoNT-A, with the most common being dysphagia (9.1%), muscle weakness (9.1%), neck pain (8.0%), and headache (5.7%); in the aboBoNT-A 500 U/1-mL pivotal trial the most common AEs were dysphagia (9%), neck pain (5%), injection site pain (5%), headache (4%), and upper respiratory tract infection (4%) [4].

Limitations of the study include the use of a fixed dose of aboBoNT-A, which is an important aspect and was done to demonstrate how BoNT-A naïve patients would respond to a standard starting dose. This dosing method differs from that used in clinical practice, where doses are individualized for each patient, and it is likely that many of the participants did not receive the optimal dose in the appropriate muscle. Nonetheless, the use of a fixed dose was necessary, as there were no data available on which to base dose optimization in naïve patients. Additional limitations of this study include the study design allowing patients who were deemed eligible to enter into the open-label extension study between Week 4 and Week 8 (before they reached the planned Week 12 end of the study visit), which resulted in 25 patients withdrawing to enter the open-label extension.

The data reported here indicate that treatment with aboBoNT-A 500 U/2-mL dilution provided patient-perceived benefits through 12 weeks. AboBoNT-A–treated patients experienced decreased pain, decreased depression, and improved health—factors associated with increased quality of life in CD patients. Furthermore, PROs were maintained over time, in line with the improved disease state. These positive outcomes support the administration of aboBoNT-A 500 U using a 2-mL dilution.

## Supporting information

**S1 Fig. Study site locations.**
(DOCX)

**S2 Fig.** Mean TWSTRS sub-scale scores across time points (ITT population): A) Mean TWSTRS severity score across time points. B) Mean TWSTRS pain score across time points. C) Mean TWSTRS disability score across time points. Tertiary endpoints: no formal statistical testing was conducted. ITT = intent-to-treat (all randomized patients); last available = last available post-baseline* (end of study or early withdrawal); TWSTRS = Toronto Western Spasmodic Torticollis Rating Scale. Error bars represent the 95% confidence interval. *Last available = last available post-baseline (end of study or early withdrawal), mean (SD) study drug exposure: 62.6 (37.8) days.
(PNG)

**S3 Fig.** Patient Global Impression of Change at (A) Week 2, (B) Week 4, and (C) last available (ITT). A. Week 2. B. Week 4. C. Last available. *Last available = last available post-baseline (end of study or early withdrawal), mean (SD) study drug exposure: 62.6 (37.8) days.
(PNG)

**S1 File.**
(PDF)

## Acknowledgments

The authors thank all patients involved in the study, as well as investigators and research staff in participating institutions, and Jim Otto, for his substantial contributions to the study while he was employed at Ipsen. The authors also thank Nicole Coolbaugh, BSc, CMPP, Kate Katsaval, BSc, and Philip Sjostedt, BPharm, of The Medicine Group, LLC (New Hope, PA, USA) for providing medical writing support.

## Author Contributions

**Conceptualization:** Atul T. Patel, Mark F. Lew, Khashayar Dashtipour, Stuart Isaacson, Robert A. Hauser, William Ondo, Pascal Maisonobe, Stefan Wietek, Bruce Rubin, Allison Brashear.

**Formal analysis:** Pascal Maisonobe.

**Investigation:** Atul T. Patel, Mark F. Lew, Khashayar Dashtipour, Stuart Isaacson, Robert A. Hauser, William Ondo, Pascal Maisonobe, Stefan Wietek, Bruce Rubin, Allison Brashear.

**Methodology:** Atul T. Patel, Mark F. Lew, Khashayar Dashtipour, Stuart Isaacson, Robert A. Hauser, William Ondo, Pascal Maisonobe, Stefan Wietek, Bruce Rubin, Allison Brashear.

**Writing – original draft:** Atul T. Patel, Mark F. Lew, Khashayar Dashtipour, Stuart Isaacson, Robert A. Hauser, William Ondo, Pascal Maisonobe, Stefan Wietek, Bruce Rubin, Allison Brashear.

**Writing – review & editing:** Atul T. Patel, Mark F. Lew, Khashayar Dashtipour, Stuart Isaacson, Robert A. Hauser, William Ondo, Pascal Maisonobe, Stefan Wietek, Bruce Rubin, Allison Brashear.

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
