## [Decision Letter · Decision Letter 0]

28 Sep 2020

PONE-D-19-35636

Sustained functional benefits after a single set of injections with abobotulinumtoxinA using a 2-mL injection volume in adults with cervical dystonia: 12-week results from a randomized, double-blind, placebo-controlled phase 3b study

PLOS ONE

Dear Dr. Wietek,

Thank you for submitting your manuscript to PLOS ONE. After careful consideration, we feel that it has merit but does not fully meet PLOS ONE’s publication criteria as it currently stands. Therefore, we invite you to submit a revised version of the manuscript that addresses the points raised during the review process.

Please respond thoroughly to all comments from the reviewers paying particular attention to expand the discussion of study limitations and how the study advances science beyond the 4 week data as required by the publication criteria (https://journals.plos.org/plosone/s/criteria-for-publication#loc-4)

We look forward to receiving your revised manuscript.

Kind regards,

Nancy Beam, PhD

Staff Editor

PLOS ONE

Journal Requirements:

a) Did participants provide their written or verbal informed consent to participate in this study?

3. Thank you for including your ethics statement:  "This study obtained appropriate institutional review board approval and was conducted under the provisions of the Declaration of Helsinki. Subjects were given a full explanation, in lay terms, of the aims of the study, the benefits, potential discomforts and risks of taking part in the study prior to enrolment. At this visit they were also given a written explanation of the study. Patients provided informed consent prior to their enrollment. Additional details are available upon request.".   

4. Thank you for providing the following Funding Statement: 

'This was funded by Ipsen, Cambridge, MA, USA (https://www.ipsen.com/us/). Grant number & authors who received the awards are not applicable. The funders had no role in study design, data collection and analysis, decision to publish, or preparation of the manuscript.'

We note that one or more of the authors is affiliated with the funding organization, indicating the funder may have had some role in the design, data collection, analysis or preparation of your manuscript for publication; in other words, the funder played an indirect role through the participation of the co-authors.

a. If the funding organization did not play a role in the study design, data collection and analysis, decision to publish, or preparation of the manuscript and only provided financial support in the form of authors' salaries and/or research materials, please review your statements relating to the author contributions, and ensure you have specifically and accurately indicated the role(s) that these authors had in your study in the Author Contributions section of the online submission form. Please make any necessary amendments directly within this section of the online submission form. 

Please also update your Funding Statement to include the following statement: “The funder provided support in the form of salaries for authors [insert relevant initials], but did not have any additional role in the study design, data collection and analysis, decision to publish, or preparation of the manuscript. The specific roles of these authors are articulated in the ‘author contributions’ section.”

If the funding organization did have an additional role, please state and explain that role within your Funding Statement.

Within your Competing Interests Statement, please confirm that this commercial affiliation does not alter your adherence to all PLOS ONE policies on sharing data and materials by including the following statement: "This does not alter our adherence to  PLOS ONE policies on sharing data and materials.” (as detailed online in our guide for authors http://journals.plos.org/plosone/s/competing-interests)

If this adherence statement is not accurate and  there are restrictions on sharing of data and/or materials, please state these.

Please note that we cannot proceed with consideration of your article until this information has been declared.

Reviewers' comments:

Reviewer's Responses to Questions

**Comments to the Author**

1. Is the manuscript technically sound, and do the data support the conclusions?

Reviewer #1: Yes

Reviewer #2: Yes

Reviewer #3: Yes

Reviewer #4: Yes

Reviewer #5: No

2. Has the statistical analysis been performed appropriately and rigorously? 

Reviewer #1: Yes

Reviewer #2: Yes

Reviewer #3: Yes

Reviewer #4: Yes

Reviewer #5: No

3. Have the authors made all data underlying the findings in their manuscript fully available?

Reviewer #1: Yes

Reviewer #2: Yes

Reviewer #3: Yes

Reviewer #4: Yes

Reviewer #5: No

4. Is the manuscript presented in an intelligible fashion and written in standard English?

Reviewer #1: No

Reviewer #2: Yes

Reviewer #3: Yes

Reviewer #4: Yes

Reviewer #5: Yes

5. Review Comments to the Author

Reviewer #1: In this study, the authors present findings on the use of abobotulinum toxin for cervical dystonia when 2ml dilution is used for preparation. The authors present data for outcomes at 12 weeks. They have already published the outcomes at four weeks follow-up. In their analysis of primary outcomes, the authors found that active group continued to have significant benefits at 12 weeks follow-up compared to the placebo. These findings indicate there are longer than expected benefits as most effects related to botulinum are expected wane at 12 weeks follow-up.

The discussion section is quite underdeveloped. Are these 12 week-outcomes different from those seen with 1ml dilution when evaluated at a similar interval of follow-up? The authors draw comparisons between 1 ml and 2 ml dilution for four weeks' follow-up. The authors should provide scientific insights to understand the value of their findings.

Reviewer #2: Last line of abstract: if most patients were measured at 12 weeks, this would be better as “at 12 weeks” rather than “up to 12 weeks”.

Pg 14- “treatment duration was between 4 and 16 weeks” - wasn’t it a single injection?

Reviewer #3: This study compares efficacy, patient-reported outcomes, and safety in CD patients receiving 2-ml aboBoNT-A or 2-ml placebo. It is shown that a treatment with 2-ml aboBoNT-A is superior compared to placebo. Besides, the results of the actual study are compared with the results of a previous 1-ml aboBoNT-A study. Since the results are comparable, the administration of aboBoNT-A 500 U using a 2-ml dilution is recommended.

My main criticism of the study is that the design and the limitations of this study have to be discussed.

Comments in detail:

Abstract:

It is only the dilution (2 ml) mentioned but not the diluted units. Please add so that the concentration is clear.

Introduction:

(page 7, line 6): ‘AbobotulinumtoxinA (aboBoNT-A; Dysport®) inhibits acetylcholine release and binds to receptors on peripheral cholinergic nerve endings, and is internalized by receptor-mediated endocytosis. This is followed by cleavage of SNAP25, a protein needed for vesicle fusion to the presynaptic membrane, which is required for synaptosomal release of neurotransmitter into the neuromuscular junction.’

This part describes the mode of action of botulinum neurotoxin in general and not of aboBoNT-A in special. So, I would recommend to change in line 6 from ‘AbobotulinumtoxinA’ to botulinum neurotoxin. Besides, the first sentence starts with the final result of the effect of BoNT-A and then goes further to the single steps of the mode of action. I would recommend to delete the first part of the sentence and to start with ‘Botulinum neurotoxin binds to…’ and change in the end of the following sentence from ‘neurotransmitter’ to ‘acetylcholine’.

Last sentence of first paragraph: AboBoNT-A is not only approved in the USA but also in Europe for cervical dystonia. This should be also mentioned.

Second paragraph: This part is a little confusing because only the volume but not the diluted BoNT-A dosis is mentioned. So, the concentration remains unclear. Since there are different vials (300 U and 500 U available) the U per ml has to be mentioned.

Methods

Treatment (line 7): oba should be changed to ona

Treatment: Regarding your study protocol the onaBoNT-A dosis was converted to aboBoNT-A dosis using a factor of 2.5. This means that the maximal dosis of onaBoNT-A was 200 U since the maximal dosis of aboBoNT-A in your study is limited to 500 U. I would like to know what happened with patients that received more than 200 U of onaBoNT-A (in table 1 it is stated that mean dose of onaBoNT-A was 177+/-34 respectively 176 +/- 42 U. So, in both treatment groups there were onaBoNT-A treated patients that received more than 200 U onaBoNT/A). Was the calculated aboBoNT-A dosis (factor 2.5:1) reduced or have these patients more than 500 U aboBoNT-A received? Please state in the Methods what you have done in such a case.

On the other hand, all BoNT-A native patients received 500 U aboBoNT-A. It is stated that ‘the sites and the dose per site were determined by the investigator according to the standard practice and disease presentation.’ But the total doses were given by the study protocol. I find this difficult because not every patient needs 500 U (corresponding to your patients that were treated with onaBoNT-A before). More explanations to the clinical usage of the fix total dose regiment in your study would be helpful.

Results

Figures: there are too many pixels. It is not possible to zoom in and to read the figures easily. Especially in figure 3B: the headings are to bold. It is not possible to read them.

Please improve.

Discussion

Third paragraph: ‘TWSTERS’ should be changed to ‘TWSTRS’

I think it is very important that a paragraph with the limitations of the study is added.

It should be discussed why only BoNT-A native patients or onaBoNT-A treated patients were included in the study. Why did you not include 1-ml aboBoNT-A treated patients and changed to a 2-ml aboBoNT-A treatment? Besides, the direct comparison between 1-ml and 2-ml aboBoNT-A treatment would be interesting. This would be more interesting than the comparison of effects of BoNT-A versus placebo. But you have compared the results of two different studies (1-ml versus 2-ml compared to placebo, respectively). So, it is not surprising that two different studies show comparable effects. It would be interesting to investigate this in one study. The result of your study that BoNT-A is the better treatment compared to placebo is not new. Please discuss this methodological difficulty of your paper.

Altogether it is not very surprising that CD patients improve when they are injected with BoNT-A compared to placebo. Patients with BoNT-A receive the ‘primary treatment for CD’ as you have stated in your introduction. It is known from literature that there are only little to no differences when the volume of BoNT-A is only doubled. To reach a volume effect the volume has to be increased much higher. Please discuss with the current literature.

Reviewer #4: This is a randomised, double-blind, placebo-controlled trial on sustained functional benefits after a single set of injections with abobotulinumtoxinA using a 2-mL injection volume in adults with cervical dystonia.

The aim of the authors was to assess the efficacy, patient-reported outcomes (PROs), and safety in adults with CD after a single set of abobotulinumtoxinA (aboBoNT-A) injections versus placebo after a single set of injections using 2-mL injection volume. 134 patients were randomised and received aboBoNT-A 500 units (U) if toxin-naïve, and 250 to 500 U based on previous onabotulinumtoxinA dose if non-naïve patients.

The conclusions were that there was a sustained improvement in TWSTRS score and patient-perceived benefits up to 12 weeks with abobotulinumtoxinA group. There was also improvement in treatment satisfaction, pain, depression, and global health.

In general, this is a complete and well written about sustained abobotulinumtoxinA efficacy in cervical dystonia.

Reviewer #5: Overall impression: I found this paper unclear to read. It is also too long for a very simple conclusion that a 2ml dilution also works. I am not convinced that this small conclusion requires a rehash of the entire study that has already been published. In my mind this is a short report only. Much of the actual 12week data is presented subjectively with very mixed if any statistics, lots of drops outs, and early treatments etc and is truly hard to interpret. I would strongly suggest the authors to substantially shorten the paper, focus on the main findings and then just present those for he 12 week follow-up. The rest of the presentation over extends the conclusion that the authors are trying to present.

In essence I don’t feel that the observations require a full and large paper.

1. Abstract: It is unclear that the patients that were non-naïve were on ona. What about inco?

2. Since all of the 4 week data is already published, this is then simply a report of what happened at 12 weeks in terms of efficacy.

3. The issue of who got placebo is totally unclear in the description. Reading the methods, it seems that the non-naïve and the naïve just got abo. No mention is made of who and how many got the placebo from which group.

4. Seems that the non-naïve patients also got placebo? That doesn’t make any sense to me. Were these patients getting efficacy with their toxin? Otr were they selected on the basis that they were not getting efficacy? If they were, they still agreed to get placebo? Is that a reason why many wanted injection earlier in the placebo group?

5. That said, the results say that there was a 40% placebo response?

6. It seems that the two cycles that were selected for ona were not the most recent but two out of 6. Presumably all of the non-naïve patients were already stable on their dose? Did patients have different injections through the 6 cycles prior to the conversion?

7. Assessments: it clearly states the endpoint is 4 weeks. Isn’t this study reporting 12 weeks data? It is not mentioned anywhere in the assessments.

8. When patients were converted, were the sites of injection for ona-versus abo kept the same? Was EMG used for both; i.e. the patients that were getting ona and then got abo had EMG for both? This will potentially make a difference in the outcomes. For example if EMG was not used for Ona and was used for Abo then that could have an outcome effect.

9. Results:

a. It is very odd that there was a 36% drop out upto week 12. Why is this so high? I presume that a number of these patients were getting Ona and then were given placebo? This seems like a poor study design to me if that is the case.

b. A similar concern is regarding the 25 Abo patients requiring earlier toxin injection than 12 weeks. Prior studies have suggested that Abo lasts longer. This casts doubt on such observations

c. Table 1 gives the details that say how many of the different toxins patients were on. 5 patients were on rima. What was the conversion ratio here?

d. It really is also confusing to report week 2 and 4 data that presumably was not the point of this 12 week study. Has this not been reported already in the already published study? For example for the PROs there is no statistics done for the week 12 and the numerical data is all for week 2 and 4. All we are given for the actual week 12 is trends

e. Even with the last known exposure, the average time is 37.8 days and not even close to 12 weeks.

10. Hence it is really quite difficult to agree with the conclusions. The data is very unclear and subjective regarding this endpoint.

11. The 12 week assessment was clearly not an a priori endpoint and that is obvious in the way the data plays out.

6. PLOS authors have the option to publish the peer review history of their article (what does this mean?). If published, this will include your full peer review and any attached files.

Reviewer #1: No

Reviewer #2: No

Reviewer #3: No

Reviewer #4: No

Reviewer #5: No

---

## [Author Response · Author response to Decision Letter 0]

21 Dec 2020

Title: Sustained functional benefits after a single set of injections with abobotulinumtoxinA using a 2-mL injection volume in adults with cervical dystonia: 12-week results from a randomized, double-blind, placebo-controlled phase 3b study (PONE-D-19-35636) PLOS ONE

Please respond thoroughly to all comments from the reviewers paying particular attention to expand the discussion of study limitations and how the study advances science beyond the 4 week data as required by the publication criteria (https://journals.plos.org/plosone/s/criteria-for-publication#loc-4)

Journal Requirements:

1. Please ensure that your manuscript meets PLOS ONE's style requirements, including those for file naming. The PLOS ONE style templates can be found at:

Author Response: Amendments to style and file have been made as needed.

a) Did participants provide their written or verbal informed consent to participate in this study?

Author Response: The corresponding paragraph in the Methods section has been expanded as follows: “Subjects were given a full explanation, in lay terms, of the aims of the study, the benefits, potential discomforts and risks of taking part in the study prior to enrollment. A written explanation was also provided and written informed consent was obtained prior to enrollment. The study protocol, subject information leaflet and informed consent document were reviewed and approved by an Institutional Review Board (IRB) prior to commencement of the study. This study was conducted under the provisions of the Declaration of Helsinki, informed consent regulations and in accordance with the International Conference on Harmonisation (ICH) Consolidated Guideline on Good Clinical Practice (GCP) and local site review boards.” (p 7, lines 143-150)

3. Thank you for including your ethics statement: "This study obtained appropriate institutional review board approval and was conducted under the provisions of the Declaration of Helsinki. Subjects were given a full explanation, in lay terms, of the aims of the study, the benefits, potential discomforts and risks of taking part in the study prior to enrolment. At this visit they were also given a written explanation of the study. Patients provided informed consent prior to their enrollment. Additional details are available upon request.". 

 Author Response: Please see author response to Question 2 above. (p 7, lines 143-150)

Author Response: Study data include potentially identifying and sensitive information. As a consequence, it is Ipsen policy to share data only with qualified researchers who provide a valid research question. Proposals for research for which research investigators wish to use an Ipsen dataset should be sent to DataSharing@Ipsen.com.

4. Thank you for providing the following Funding Statement: 

We note that one or more of the authors is affiliated with the funding organization, indicating the funder may have had some role in the design, data collection, analysis or preparation of your manuscript for publication; in other words, the funder played an indirect role through the participation of the co-authors.

a. If the funding organization did not play a role in the study design, data collection and analysis, decision to publish, or preparation of the manuscript and only provided financial support in the form of authors' salaries and/or research materials, please review your statements relating to the author contributions, and ensure you have specifically and accurately indicated the role(s) that these authors had in your study in the Author Contributions section of the online submission form. Please make any necessary amendments directly within this section of the online submission form. 

Please also update your Funding Statement to include the following statement: “The funder provided support in the form of salaries for authors [insert relevant initials], but did not have any additional role in the study design, data collection and analysis, decision to publish, or preparation of the manuscript. The specific roles of these authors are articulated in the ‘author contributions’ section.”

If the funding organization did have an additional role, please state and explain that role within your Funding Statement.

Within your Competing Interests Statement, please confirm that this commercial affiliation does not alter your adherence to all PLOS ONE policies on sharing data and materials by including the following statement: "This does not alter our adherence to PLOS ONE policies on sharing data and materials.” (as detailed online in our guide for authors http://journals.plos.org/plosone/s/competing-interests)

If this adherence statement is not accurate and there are restrictions on sharing of data and/or materials, please state these.

Please note that we cannot proceed with consideration of your article until this information has been declared.

Author Response: The following information has been added to the ‘Funding/Sponsor’ section: “The funder provided support in the form of salaries for authors [P.M., S.W., and B.R. (formerly)] and support for medical writing of the manuscript, but did not have any additional role in the study design, data collection and analysis, or decision to publish. The specific roles of these authors are articulated in the ‘author contributions’ section.” (p 2, lines 33-37)

Reviewer's Responses to Questions

We thank the reviewers for their feedback and appreciate the opportunity to respond.

Reviewer #1: 

1. In this study, the authors present findings on the use of abobotulinum toxin for cervical dystonia when 2ml dilution is used for preparation. The authors present data for outcomes at 12 weeks. They have already published the outcomes at four weeks follow-up. In their analysis of primary outcomes, the authors found that the active group continued to have significant benefits at 12 weeks follow-up compared to placebo. These findings indicate there are longer than expected benefits, as most effects related to botulinum are expected to wane at 12 weeks follow-up.

2. The discussion section is quite underdeveloped. Are these 12 week-outcomes different from those seen with 1ml dilution when evaluated at a similar interval of follow-up? The authors draw comparisons between 1 ml and 2 ml dilution for four weeks' follow-up. The authors should provide scientific insights to understand the value of their findings.

Author Response: The goal of this study was to present the efficacy and safety of aboBoNT-A 500 U/2 mL versus placebo, for up to 12 weeks. These 12-week outcomes are in fact similar to the results seen with the 1 mL dilution, further supporting the claim that using a 500 U/2 mL dilution of aboBoNT-A is just as safe and effective as the 1 mL dilution. These results not only provide scientific insight into the efficacy of aboBoNT-A using a 2 mL dilution, but also the factors associated with increased quality of life.

We have added the following information to the Discussion section: “The weighted overall treatment difference between aboBoNT-A and placebo was –4.8 at Week 12 or at early study withdrawal in the current trial, similar to the adjusted mean difference of –4.3 and -4.1 in the pivotal clinical trials at Week 12.” (p 20, 406-409)

Reviewer #2: 

1. Last line of abstract: if most patients were measured at 12 weeks, this would be better as “at 12 weeks” rather than “up to 12 weeks”.

Author Response: We strongly believe that keeping the last line of the abstract as is, “up to 12 weeks” is more accurate due to the fact that patients who withdrew prior to the end of the study visit at week 12 still underwent all procedures required to the Week 12 visit. Therefore, keeping the statement as is captures the fact that the data are representative of all patients, those who completed the Week 12 visit, as well as those who withdrew from the study early.

2. Pg 14- “treatment duration was between 4 and 16 weeks” - wasn’t it a single injection?

Author Response: This study had the following visits, pre-study screening (Day −7 to Day −1, Visit 1), baseline (Day 1, Visit 2), post-treatment follow up (Week 2±2 days, Visit 3 and Week 4±2 days, Visit 4) and study completion or early withdrawal (Week 12±28 days, Visit 5). Patients were given an extra 2-day window for their Visit 3 and 4, and a 28-day window for Visit 5, resulting in the potential for some patients to be enrolled in the study for up to 16 weeks.

We have now expanded the corresponding sentence in the Results section to clarify the reason for treatment duration variation from 4 to 16 weeks, as follows: “Individual treatment duration was between 4 weeks and 16 weeks, due to the study design, which allowed a ±2 day window to complete the Week 2 and 4 visits, and a 28 day window to complete the Week 12 visit.” (p 13, lines 278-280)

Reviewer #3: 

This study compares efficacy, patient-reported outcomes, and safety in CD patients receiving 2-ml aboBoNT-A or 2-ml placebo. It is shown that a treatment with 2-ml aboBoNT-A is superior compared to placebo. Besides, the results of the actual study are compared with the results of a previous 1-ml aboBoNT-A study. Since the results are comparable, the administration of aboBoNT-A 500 U using a 2-ml dilution is recommended.

My main criticism of the study is that the design and the limitations of this study have to be discussed.

Comments in detail:

1. Abstract: It is only the dilution (2 ml) mentioned but not the diluted units. Please add so that the concentration is clear.

Author Response: The units using 2-mL dilution have now been added, as follows: “We present efficacy, patient-reported outcomes (PROs), and safety in adults with CD at the last available visit after a single set of abobotulinumtoxinA (aboBoNT-A) injections versus placebo using 500 U, 2-mL injection volume” (p 5, line 88)

2. Introduction: (page 7, line 6): ‘AbobotulinumtoxinA (aboBoNT-A; Dysport®) inhibits acetylcholine release and binds to receptors on peripheral cholinergic nerve endings, and is internalized by receptor-mediated endocytosis. This is followed by cleavage of SNAP25, a protein needed for vesicle fusion to the presynaptic membrane, which is required for synaptosomal release of neurotransmitter into the neuromuscular junction.’

This part describes the mode of action of botulinum neurotoxin in general and not of aboBoNT-A in special. So, I would recommend to change in line 6 from ‘AbobotulinumtoxinA’ to botulinum neurotoxin. Besides, the first sentence starts with the final result of the effect of BoNT-A and then goes further to the single steps of the mode of action. I would recommend to delete the first part of the sentence and to start with ‘Botulinum neurotoxin binds to…’ and change in the end of the following sentence from ‘neurotransmitter’ to ‘acetylcholine’.

Author Response: The corresponding sentence in the Introduction has now been amended to the following, as suggested by the reviewer: “Botulinum neurotoxin type A binds to receptors on peripheral cholinergic nerve endings and is internalized by receptor-mediated endocytosis.” (p 6, lines 114-116, 118-119)

3. Last sentence of first paragraph: AboBoNT-A is not only approved in the USA but also in Europe for cervical dystonia. This should be also mentioned.

Author Response: The corresponding sentence in the Introduction has now been amended to the following, as suggested by the reviewer: “AboBoNT-A is approved in the United States of America (US) and Europe for the treatment of adults with CD” (p 6, lines 120-121)

4. Second paragraph: This part is a little confusing because only the volume but not the diluted BoNT-A dosis is mentioned. So, the concentration remains unclear. Since there are different vials (300 U and 500 U available) the U per ml has to be mentioned.

Author Response: The sentence has now been expanded as follows: “Initially, the prescribing information in the US specified a 500-unit (U), 1-mL dilution regimen for AboBoNT-A administration; however, feedback obtained from scientific experts, community injectors, and market research studies favored use of a 500 U, 2-mL dilution.” (p 6, lines 123-125)

5. Methods, Treatment (line 7): oba should be changed to ona

Author Response: This correction has been made. (p 9, line 177)

6. Treatment: 

a. Regarding your study protocol the onaBoNT-A dosis was converted to aboBoNT-A dosis using a factor of 2.5. This means that the maximal dosis of onaBoNT-A was 200 U since the maximal dosis of aboBoNT-A in your study is limited to 500 U. I would like to know what happened with patients that received more than 200 U of onaBoNT-A (in table 1 it is stated that mean dose of onaBoNT-A was 177+/-34 respectively 176 +/- 42 U. So, in both treatment groups there were onaBoNT-A treated patients that received more than 200 U onaBoNT/A). Was the calculated aboBoNT-A dosis (factor 2.5:1) reduced or have these patients more than 500 U aboBoNT-A received? Please state in the Methods what you have done in such a case.

Author Response: Although the eligibility criteria for this study required that if patients were currently being treated with onaBoNT-A, their dose did not exceed 200 U, 17 patients (10 in the aboBoNT-A group, and 7 in the placebo group) were excluded from the efficacy analysis in the intent-to-treat population due to eligibility criteria violations. As a result, the mean dose of the most recent treatment with onaBoNT-A in both the aboBoNT-A group and in the placebo group does exceed 200 U. These patients, however, were excluded and did not receive the study treatment drug. The footer in Table 1 has been updated to clarify why the mean dose of onaBoNT-A previously received exceeded study eligibility criteria. (p 14, lines 285-287)

b. On the other hand, all BoNT-A native patients received 500 U aboBoNT-A. It is stated that ‘the sites and the dose per site were determined by the investigator according to the standard practice and disease presentation.’ But the total doses were given by the study protocol. I find this difficult because not every patient needs 500 U (corresponding to your patients that were treated with onaBoNT-A before). More explanations to the clinical usage of the fix total dose regiment in your study would be helpful.

Author Response: All BoNT-A naïve patients received 500 U/2-mL of aboBoNT-A based on the results of a previous study (Poewe, et al. 1998) which found the optimal dose of aboBoNT-A for patients with cervical dystonia to be 500 U. We believe this to be a limitation of the study as this dosing procedure differs from that used in clinical practice, where the doses are individualized per patient. Nevertheless, the use of a fixed dose was deemed necessary, as there were no data on which to base dose optimization in BoNT-A naïve patients. This information has now been added to a ‘limitations’ paragraph in the manuscript Discussion section. (p 21, lines 417-426)

7. Results, Figures: there are too many pixels. It is not possible to zoom in and to read the figures easily. Especially in figure 3B: the headings are to bold. It is not possible to read them. Please improve.

Author Response: Resolution of the Figures has been updated for clarity. 

8. Discussion, Third paragraph: ‘TWSTERS’ should be changed to ‘TWSTRS’. 

Author Response: this correction has now been made. (p 20, line 404)

a. I think it is very important that a paragraph with the limitations of the study is added.

It should be discussed why only BoNT-A native patients or onaBoNT-A treated patients were included in the study. Why did you not include 1-ml aboBoNT-A treated patients and changed to a 2-ml aboBoNT-A treatment? Besides, the direct comparison between 1-ml and 2-ml aboBoNT-A treatment would be interesting. This would be more interesting than the comparison of effects of BoNT-A versus placebo. But you have compared the results of two different studies (1-ml versus 2-ml compared to placebo, respectively). So, it is not surprising that two different studies show comparable effects. It would be interesting to investigate this in one study. The result of your study that BoNT-A is the better treatment compared to placebo is not new. Please discuss this methodological difficulty of your paper.

Altogether it is not very surprising that CD patients improve when they are injected with BoNT-A compared to placebo. Patients with BoNT-A receive the ‘primary treatment for CD’ as you have stated in your introduction. It is known from literature that there are only little to no differences when the volume of BoNT-A is only doubled. To reach a volume effect the volume has to be increased much higher. Please discuss with the current literature.

Author Response: The following text has now been added: “Limitations of the study include the use of a fixed dose of aboBoNT-A, which is an important aspect and was done to demonstrate how BoNT-A naïve patients would respond to a standard starting dose. This dosing method differs from that used in clinical practice, where doses are individualized for each patient, and it is likely that many of the participants did not receive the optimal dose in the appropriate muscle. Nonetheless, the use of a fixed dose was necessary, as there were no data available on which to base dose optimization in naïve patients. Additional limitations of this study include the study design allowing patients who were deemed eligible to enter into the open-label extension study between Week 4 and Week 8 (before they reached the planned Week 12 end of the study visit), which resulted in 25 patients withdrawing from the study to enter the open-label extension.” (p 21, lines 417-426)

We agree that a direct comparison of a 1 mL and 2 mL dilution to placebo would be interesting, and we hope based on the results of our study, it can be done in the future.

Although the efficacy of aboBoNT-A versus placebo is not new, safety and efficacy of a single 500 U/2 mL dilution injection of aboBoNT-A for up to 12 weeks has not yet been demonstrated, as it has been with a 1 mL dilution. As there had been a lack of scientific data and literature regarding the use of a 500 U/2 mL dilution of aboBoNT-A versus placebo, we believe that the results of this study support the efficacy and safety of using a 500 U/2 mL dilution to better meet patient needs. 

Feedback received from scientific experts, community injectors, and market research studies favored the use of a 2 mL dilution, specifically. This was most likely related to prior familiarity in using comparable volumes with other approved toxins in the US. In a study conducted in Europe (Yun, et al. 2015), a 2 mL dilution of aboBoNT-A was used to demonstrate the efficacy and safety of aboBoNT-A in comparison with onaBoNT-A. Therefore, in this clinical study, the majority of enrolled patients had previously been treated with onaBoNT-A, in order to reflect the real world clinical scenario in the US.

Although different indications may allow larger dilution volumes, the recommended concentration range for cervical dystonia is 25 or 50 U/0.1 mL, with a aboBoNT-A dose of 500 to 1000 U. (No changes to the manuscript text)

Reviewer #4: This is a randomised, double-blind, placebo-controlled trial on sustained functional benefits after a single set of injections with abobotulinumtoxinA using a 2-mL injection volume in adults with cervical dystonia.

The aim of the authors was to assess the efficacy, patient-reported outcomes (PROs), and safety in adults with CD after a single set of abobotulinumtoxinA (aboBoNT-A) injections versus placebo after a single set of injections using 2-mL injection volume. 134 patients were randomised and received aboBoNT-A 500 units (U) if toxin-naïve, and 250 to 500 U based on previous onabotulinumtoxinA dose if non-naïve patients.

The conclusions were that there was a sustained improvement in TWSTRS score and patient-perceived benefits up to 12 weeks with abobotulinumtoxinA group. There was also improvement in treatment satisfaction, pain, depression, and global health.

In general, this is a complete and well written about sustained abobotulinumtoxinA efficacy in cervical dystonia.

 Author Response: We thank Reviewer #4 for this review and comments.

Reviewer #5: 

Overall impression: I found this paper unclear to read. It is also too long for a very simple conclusion that a 2ml dilution also works. I am not convinced that this small conclusion requires a rehash of the entire study that has already been published. In my mind this is a short report only. Much of the actual 12week data is presented subjectively with very mixed if any statistics, lots of drops outs, and early treatments etc and is truly hard to interpret. I would strongly suggest the authors to substantially shorten the paper, focus on the main findings and then just present those for the 12 week follow-up. The rest of the presentation over extends the conclusion that the authors are trying to present. In essence I don’t feel that the observations require a full and large paper.

a. Abstract: It is unclear that the patients that were non-naïve were on ona. What about inco?

Author Response: As previously published in the primary manuscript, eligible patients who were non-naïve and currently undergoing treatment with onaBoNT-A (receiving a total dose of 100-200 U, and ≤60 U in the sternocleidomastoid muscle (SCM), since the last injection cycle, could have received any other formulation of BoNT-A prior to study enrollment, as long as they had a satisfactory clinical response to the last two sequential cycles with onaBoNT-A in the past 18 months, or had not received onaBoNT-A for at least 12 weeks. (No changes to manuscript text)

b. Since all of the 4-week data is already published, this is then simply a report of what happened at 12 weeks in terms of efficacy.

Author Response: This study presents the efficacy, patient-reported outcomes, and safety results at the last available visit, after a single injection of aboBoNT-A 500 U/2 mL dilution versus placebo, further supporting the results of the primary publication. 

As the methodology and primary efficacy endpoints have already been published, we have included only the necessary data. 

The study rationale was based on previously published data (Yun, et al, 2015) which demonstrated that aboBoNT-A at a conversion ratio of 2.5:1 was not inferior in comparison to onaBoNT-A, utilizing a 2 mL dilution. (No changes to manuscript text)

c. The issue of who got placebo is totally unclear in the description. Reading the methods, it seems that the non-naïve and the naïve just got abo. No mention is made of who and how many got the placebo from which group.

Author Response: Patients were randomized in blocks, based on a computer-generated randomization list. List A stratified patients who were toxin naïve or non-naïve at baseline with a 2:1 ratio of aboBoNT-A:placebo. List B was produced on a 1:1 basis (aboBoNT-A:placebo) of treatment numbers, which were specified on the treatment packs, to be dispatched to the sites in order to dispense the drug. (No changes to manuscript text)

d. Seems that the non-naïve patients also got placebo? That doesn’t make any sense to me. Were these patients getting efficacy with their toxin? Otr were they selected on the basis that they were not getting efficacy? If they were, they still agreed to get placebo? Is that a reason why many wanted injection earlier in the placebo group?

Author Response: Please see the above response (Reviewer 5, comment “c”). 

e. That said, the results say that there was a 40% placebo response?

Author Response: Although 40% of patients in the placebo group reported being at least somewhat satisfied with the medication, approximately twice as many patients in the aboBoNT-A group were at least somewhat satisfied across time points (Week 2: abobotulinumtoxinA, 64.8%, vs placebo, 32.5%; Week 4: abobotulinumtoxinA, 65.1%, vs placebo, 35.6%; last available: abobotulinumtoxinA, 64.0%, vs placebo, 40.0%). (No changes to manuscript text)

f. It seems that the two cycles that were selected for ona were not the most recent but two out of 6. Presumably all of the non-naïve patients were already stable on their dose? Did patients have different injections through the 6 cycles prior to the conversion?

Author Response: Eligible non-naïve patients currently treated with onaBoNT-A at a total dosing range of 100 U to 200 U and ≤60 U in the SCM at the last injection cycle and had a satisfactory treatment response in the enrolling investigator's judgment during the last two sequential cycles of onaBoNT-A within the past 18 months for the treatment of CD, and a minimum of 12 weeks since the last onaBoNT-A injection. (No changes to manuscript text)

g. Assessments: it clearly states the endpoint is 4 weeks. Isn’t this study reporting 12 weeks data? It is not mentioned anywhere in the assessments.

Author Response: This statement was in the Statistical analysis section. It has now been added to the study design. “The study had the following visits: prestudy screening (Day -7 to Day -1, Visit 1), baseline (Day 1, Visit 2; this visit could have occurred on the same day as the screening visit at the discretion of the investigator), post-treatment follow up (Week 2±2 days, Visit 3 and Week 4±2 days, Visit 4) and study completion or early withdrawal (Week 12+28 days, Visit 5). All subjects who had completed the Week 12 visit were considered to have completed the study.”

Also added to assessments “Tertiary efficacy endpoints included assessments of the change from baseline at Week 12.” (Page 9, lines 190-191)

h. When patients were converted, were the sites of injection for ona-versus abo kept the same? Was EMG used for both; i.e. the patients that were getting ona and then got abo had EMG for both? This will potentially make a difference in the outcomes. For example if EMG was not used for Ona and was used for Abo then that could have an outcome effect.

Author Response: Study drug injections in non-naïve patients were given into previously injected muscles (Study Methods, p 8, 157-161). Electromyography-guided injections were allowed, at the preference of the investigator at each site (Treatment, p 8, lines 174-175). (No changes to manuscript text)

i. Results:

a. It is very odd that there was a 36% drop out up to week 12. Why is this so high? I presume that a number of these patients were getting Ona and then were given placebo? This seems like a poor study design to me if that is the case.

b. A similar concern is regarding the 25 Abo patients requiring earlier toxin injection than 12 weeks. Prior studies have suggested that Abo lasts longer. This casts doubt on such observations

Author Response: Reasons for patient exclusions from the study were eligibility criteria violations, randomization/treatment allocation process violations, study treatment administration violation, time window violations, and TWSTRS data not available, or the patient was not treated.

Due to the study design, eligible patients were allowed to enter into the open-label extension study between Week 4 and Week 8, and prior to the Week 12 study conclusion visit. As a result, 47 patients (25 in aboBoNT-A, 22 in placebo) elected to enter the open-label extension portion of the study prior to Week 12, with 33 patients entering at Week 4. This information is already included in the manuscript. (No changes to manuscript text)

j. Table 1 gives the details that say how many of the different toxins patients were on. 5 patients were on rima. What was the conversion ratio here?

Author Response: Although patients receiving other formulations were included, the conversion ratio used was aboBoNT-A:onaBoNT-A. Table 1 summarizes the previous BoNT-A treatments the patients had received. (No changes to manuscript text)

k. It really is also confusing to report week 2 and 4 data that presumably was not the point of this 12 week study. Has this not been reported already in the already published study? For example for the PROs there is no statistics done for the week 12 and the numerical data is all for week 2 and 4. All we are given for the actual week 12 is trends

Author Response: Week 2 and Week 4 data are included to show the efficacy and safety, and patient-level benefits of aboBoNT-A as early as Week 2 and up to Week 12. (No changes to manuscript text)

l. Even with the last known exposure, the average time is 37.8 days and not even close to 12 weeks.

Author Response: The average study exposure, as stated in Table 2, for the total intent-to-treat population was 62.6 (SD 37.8) days, with a median of 85.0 days. In the aboBoNT-A group, the average study exposure was 67.7 (SD 38.4). (No changes to manuscript text)

m. Hence it is really quite difficult to agree with the conclusions. The data is very unclear and subjective regarding this endpoint.

Author Response: These findings do in fact support our conclusions in terms of the efficacy of aboBoNT-A having longer than expected benefits. (No changes to manuscript text)

n. The 12 week assessment was clearly not an a priori endpoint and that is obvious in the way the data plays out.

Author Response: A hierarchical testing procedure combined the primary and ranked secondary efficacy endpoints together with type I error controlled at a level of 5%. The superiority of aboBoNT-A to placebo on the rank-1 secondary efficacy endpoint was tested using the same method as the one used for the primary efficacy endpoint. The hierarchical testing procedure was stopped if there was no statistically significant treatment effect on the current efficacy endpoint, otherwise the next step was performed. Exploratory analysis was performed for each of the tertiary efficacy endpoints. Summary statistics were summarized by treatment group and not compared by formal statistical testing. Confidence intervals were estimated for some endpoints to characterize the full clinical effect but no statistical conclusions were drawn. (No changes to manuscript text)

6. PLOS authors have the option to publish the peer review history of their article (what does this mean?). If published, this will include your full peer review and any attached files.

Do you want your identity to be public for this peer review? For information about this choice, including consent withdrawal, please see our Privacy Policy.

Reviewer #1: No

Reviewer #2: No

Reviewer #3: No

Reviewer #4: No

Reviewer #5: No

---

## [Editor Report · Decision Letter 1]

11 Jan 2021

Sustained functional benefits after a single set of injections with abobotulinumtoxinA using a 2-mL injection volume in adults with cervical dystonia: 12-week results from a randomized, double-blind, placebo-controlled phase 3b study

PONE-D-19-35636R1

Dear Dr. Wietek,

We’re pleased to inform you that your manuscript has been judged scientifically suitable for publication and will be formally accepted for publication once it meets all outstanding technical requirements.

Kind regards,

Mandar S Jog

Guest Editor

PLOS ONE
---

## [Editor Report · Acceptance letter]

19 Jan 2021

PONE-D-19-35636R1 

Sustained functional benefits after a single set of injections with abobotulinumtoxinA using a 2-mL injection volume in adults with cervical dystonia: 12-week results from a randomized, double-blind, placebo-controlled phase 3b study 

Dear Dr. Wietek:

I'm pleased to inform you that your manuscript has been deemed suitable for publication in PLOS ONE. Congratulations! Your manuscript is now with our production department. 

Kind regards, 

on behalf of

Dr. Mandar S Jog 

Guest Editor

PLOS ONE